# Does Dynamic Tailoring of A Narrative-Driven Exergame Result in Higher User Engagement among Adolescents? Results from A Cluster-Randomized Controlled Trial

**DOI:** 10.3390/ijerph18147444

**Published:** 2021-07-12

**Authors:** Ayla Schwarz, Greet Cardon, Sebastien Chastin, Jeroen Stragier, Lieven De Marez, Ann DeSmet

**Affiliations:** 1Consumption and Healthy Lifestyles, Faculty of Social Sciences, Wageningen University & Research, 6708 PB Wageningen, The Netherlands; ayla.schwarz@wur.nl; 2Department of Movement and Sport Sciences, Ghent University, 9000 Ghent, Belgium; greet.cardon@ugent.be; 3School of Health and Life Science, Caledonian University, Glasgow G4 0BA, UK; Sebastien.Chastin@gcu.ac.uk; 4IMEC-MICT, Department of Communication Sciences, Ghent University, 9000 Ghent, Belgium; jeroen.stragier@ugent.be (J.S.); lieven.demarez@ugent.be (L.D.M.); 5KnowledgeBizConsulting, and Faculdade de Ciências e Tecnologia, Universidade NOVA, 1099-085 Lisbon, Portugal; Info@smartlifeproject.eu; 6Research Center for the Promotion of Health, Prosocial Behavior and Wellbeing PACE, Faculty of Psychological and Educational Sciences, Université Libre de Bruxelles, 1050 Brussels, Belgium; 7Department of Communication Studies, University of Antwerp, 2000 Antwerp, Belgium

**Keywords:** mobile exergame, serious game, user engagement, adolescent, dynamic tailoring, dynamic difficulty adjustment, randomized controlled trial

## Abstract

Physical activity interventions for youth are direly needed given low adherence to physical activity guidelines, but many interventions suffer from low user engagement. Exergames that require bodily movement while played may provide an engaging form of physical activity intervention but are not perceived as engaging to all. This study aimed to evaluate whether dynamic tailoring in a narrative-driven mobile exergame for adolescents played in leisure settings, can create higher user engagement compared to a non-tailored exergame. A cluster-randomized controlled trial assessed differences in user engagement between a dynamically tailored (based on an accelerometer sensor integrated in a T-shirt) and non-tailored condition. In total, 94 participants (M age = 14.61 ± 1.93; 35% female) participated and were assigned to one of the two conditions. User engagement was measured via a survey and game metric data. User engagement was low in both conditions. Narrative sensation was higher in the dynamically tailored condition, but the non-tailored condition showed longer play-time. User suggestions to create a more appealing game included simple and more colorful graphics, avoiding technical problems, more variety and shorter missions and multiplayer options. Less cumbersome or more attractive sensing options than the smart T-shirt may offer a more engaging solution, to be tested in future research.

## 1. Introduction

Physical activity among adolescents is an important area for health promotion: it contributes to both physical and mental health [1,2], and tracks into adulthood [3]. Physical activity interventions are needed because approximately 80 percent of adolescents worldwide currently do not meet activity level guidelines [4,5]. However, current physical activity interventions face challenges in motivating and engaging adolescents, which results in low usage, non-adherence or early drop-out [6,7,8]. Engagement reflects the involvement and motivation of the user in the intervention [9] and can be defined both in terms of subjective engagement of the experience and behavioral engagement of intervention usage patterns [10]. Serious games may help to reduce these challenges by being intrinsically motivating to play [11]. Serious games are games that are both intended for entertainment and educational purposes [12,13] and have shown to be effective in promoting a variety of health behaviors [14]. Games for physical activity promotion are often referred to as active video games or exergames, which requires body movement of the player to play the game [15]. Exergames are often preferred by youth over regular physical activity exercises [16], increase youths self-efficacy in being physically active [17] and can be effective at increasing light to moderate physical activity levels [18].

### 1.1. Game Engagement

Serious games, however, do not appear to be engaging for everyone [19,20,21]. Game engagement has been described in terms of motivation to start playing (i.e., looking for challenge, competition), in-game subjective engagement (i.e., enjoyment), game usage, game loyalty or continued usage, and its effects on the player’s quality of life [19]. In-game engagement has been further detailed as an experience of flow (i.e., enjoyment and feelings of control that stem from a balance between challenge and skill), immersion (i.e., a feeling of being part of the game environment with some awareness of one’s surroundings), presence (i.e., experiencing spatial presence in the game environment), absorption (i.e., losing track of time) [22], positive (i.e., pleasure) and negative affect (i.e., disappointment), tension (i.e., feelings of suspense and arousal), and competence (i.e., feelings of accomplishment and pride) [23].

Engagement is a precondition for health interventions to be effective: if interventions are not used, or not in the way, as long or as intensively as is needed, we cannot expect these interventions to have an effect on behavioural or clinical outcomes [10,24]. Engagement is influenced by characteristics of the intervention (e.g., content, technological features) and in its turn influences determinants of behavioural change by the depth of involvement with the intervention: an intervention that is better appreciated by users will yield higher attention and motivation to interact with the intervention components that create effects [24,25]. Understanding game engagement and what contributes to it is thus important for digital health interventions seeking to have an impact on public health.

### 1.2. Elements That Influence Game Engagement

Serious games that are played during leisure rather than in structured settings such as schools especially suffer from low engagement [26]. In a structured setting (e.g., school), a serious game is often perceived as more fun than the alternative (i.e., a didactic lesson) [27]. In leisure settings, however, serious games need to compete with commercial off-the-shelf games or other leisure time activities and can face reduced interest over time [28], boredom, or preferences for other games [29].

The lack of a narrative in exergames has been suggested as one of the reasons exergames are often experienced as boring [30]. Narratives (i.e., stories, anecdotes) refer to a series of connected events that take place in a setting and involve characters and conflict [31]. When the user feels transported into the narrative event and identifies with the characters, there will be a higher level of readiness to accept the health message embedded in the narrative [32]. Narrative-based health games were indeed perceived as more enjoyable [33].

A tailored approach that matches the content and methods to the individual needs of the user may help increase user engagement with serious games. Tailoring (also referred to as customization, adaptation or personalization) increases the personal relevance of the tool for the user and hence results in higher attention to, interest for and retention of content, and higher usage of a health intervention [34]. Higher usage rates and in-game motivation were indeed observed in tailored educational games compared to non-tailored versions [35]. A promising form of tailoring to increase user engagement in games is that of dynamic challenge tailoring, or dynamic difficulty adaptation.

### 1.3. Potential of Dynamic Tailoring to Enhance Game Engagement

Dynamic tailoring involves multiple assessments of the behavior (i.e., physical activity) and continuously creating an optimal balance between the challenge and the player’s skills and progress [36,37]. This by definition has the potential to promote flow [38], as also documented in experimental game research [39]. Moreover, such dynamic tailoring has been shown to increase play duration [40], user enjoyment, and immersion in games [41,42]. In games, static challenges were indeed associated with boredom, demotivation [43,44], and less immersion [41]. Research on dynamic tailoring in exergames in scarce. Earlier research suggests that games with higher difficulties and integrated narratives may evoke feelings of presence and proved that dynamic tailoring can contribute to higher presence [45]. A study conducted among young adults in a lab setting showed that a dynamically adapted version of a 15-min narrative-driven exergame resulted in a higher game rating, enjoyment, motivation for future play, feelings of competence, self-efficacy for physical activity, and recommending to others compared to a non-adapted version [46]. Another study conducted in a lab setting also showed that a dynamically tailored version of an exergame improved flow. Several methods of dynamic tailoring were used, with the most optimal effects found for tailoring based on biofeedback [47]. To the best of our knowledge, no study on dynamically tailored exergames has examined effects on game engagement in leisure settings.

### 1.4. Study Aims

The aim of this study was to evaluate whether dynamic tailoring of the challenge in a narrative-driven exergame, based on users’ in-game biofeedback, resulted in higher user engagement when played in leisure settings among adolescents. We hypothesized that the group whose challenge was dynamically tailored to their in-game biofeedback would experience higher levels of engagement than the non-tailored group. We especially hypothesized that we would find higher engagement rates and usage rates (gameplay duration, continued gameplay), enjoyment, immersion, presence and flow.

Biofeedback was enabled by a flat and small sensor (accelerometer) placed in a 3 × 3 cm pocket at the back of a T-shirt specifically designed for the purpose of this study. A T-shirt may offer comfort and the potential for personalization in different colors and prints, which may appeal to an adolescent target population. Battery life and washability are important issues as well that can present some manufacturing challenges for smart shirts. The choice for a sensor in the shirt rather than a fully embedded sensor in the fabric was driven by reasons of feasibility in this stage of the intervention’s pilot testing [48]. Ideally in a finalized product, the measurement would be made by conductive textile, where the sensor is not immediately visible and that appears more natural and integrated to the user. As the purpose of this paper was to assess the role of the biofeedback in user engagement, a fully embedded sensor was not considered to be needed here. Despite some increasing interest in such solutions (see e.g., [49] for a study protocol of an intervention combining a smart shirt and serious game), we are not aware of any other study assessing the effects of the combination of a smart shirt and a serious game on user engagement, which again highlights the innovativeness of this study.

This paper first describes the intervention, methods of data collection and analyses; then presents the characteristics of the analyzed sample, results of the quantitative analyses on engagement, and qualitative analyses of user comments, and ends with a discussion of these results, their contribution to the literature and practical implications. Some limitations and strengths are provided as a final note in the paper.

## 2. Materials and Methods

The study is part of the European SmartLife project (www.smartlifeproject.eu) which aims to increase physical activity among adolescents using a mobile exergame [50,51]. The game developed for this project consisted of a narrative-driven, single player, audio-supported mobile exergame. The narrative integrated in the SmartLife game may provide an enhanced engagement with the plot and result in increased replayability [52]. The story is set in a post-apocalyptic world where the player needs to perform physical activities to charge his/her protective suit and maintain the bunker where he/she stays. By performing the missions, it eventually becomes clear to players that there are other survivors, and together they can reach an uncontaminated island (Figure 1). The player needs to be physically active, especially using lower body movement in order to proceed in the game. The dynamically tailored group played a version of the game where the challenge was adapted in real time based on biofeedback on their activity levels, measured with an accelerometer sensor in a T-shirt that connects to the smartphone via Bluetooth (‘dynamically tailored group’). If the sensor detects that the player is walking slowly, the narrative will adapt and require the player to move more intensively. For example, in the game, a storm is forecast when the player is only engaging in lower intensity activities, hurrying the player to be more active to reach a shelter. The non-tailored group played a version of the game without a game T-shirt where no activity measurements were taken and used for tailoring. Apart from the presence or absence of dynamic tailoring, the game was identical. A detailed description of the game can be found elsewhere [50,51,53]. The game has been developed in close collaboration with the target group, game developers and researchers, by following a participatory game design [54]. Whereas the larger study included two intervention groups (two different game versions) and one control group (that did not receive any intervention), this article exclusively focused on the differences in user engagement with the game between the two intervention groups. The study is registered within the clinical trial registry clinical_trials.gov (ID: NCT03659604) and ethical approval was received from the Ethics Committee of Ghent University Hospital (registration: B670201836878).

### 2.1. Target Population and Sampling Methods

Adolescents aged between 12 and 18 years old were considered eligible for the study. Participants were recruited from a convenience sample of 20 secondary schools in Flanders, Belgium, that offered academic and/or non-academic track education. The schools were contacted via the personal network of the researchers (e.g., schools where master thesis students who were helping out with recruitment were alumni, which were located in the vicinity of the university, of which the researchers or colleagues knew some of the staff). Adolescents were eligible to participate if they possessed an Android smartphone (minimum Android version 5) and were not restricted in their movement by a medical condition. Consent from both parents and adolescents was required to allow participation, in line with ethical guidelines.

### 2.2. Procedure

The present study is a cluster-randomized controlled trial, comparing two intervention conditions (dynamically tailored group, non-tailored group). School classes were randomly allocated to one of the intervention groups, to avoid social biases between conditions. The game was downloaded on their smartphones at school with the support of a researcher, adolescents were asked to play the game during their leisure time for the following four weeks. After four weeks, both intervention conditions completed a survey on game engagement and exported the game metric data.

A mixed-method approach of quantitative analysis of scales and qualitative analysis of open-ended responses was chosen to combine their advantages in measuring engagement. Using standardized scales allows comparisons by user characteristics but lacks depth, whereas a qualitative analysis of individual’s experiences is time-consuming to analyze, requires particpants to understand their motives and behavior, but offers more insights and potential hypotheses to understand trends in quantitative data [24,25].

### 2.3. Measurements

#### 2.3.1. Participant Characteristics

Socio-demographic information included gender (female/male), type of education (academic/non-academic track), age and family affluence [55]. A 7-point scale assessed the frequency of playing digital games and exergames at baseline measurement: never, less than monthly, monthly, weekly, several times a week, daily, several times a day. As this variable was severely skewed, it was dichotomized as playing daily (≥daily) and playing infrequently (<daily).

#### 2.3.2. User Engagement

Behavioral engagement was measured continuously and exported once (after four weeks) based on game metric data that was saved locally on the smartphone and collected information on the amount of total play time and number of completed sessions. Subjective engagement was measured once after four weeks of playtime with the validated Kids Game Experience Questionnaire (GEQ) [23] on a 5-point scale (i.e., 0 indicating disagreement, 4 indicating agreement), including three items per construct. Cronbach’s alpha showed good to excellent internal consistency for the overall GEQ scale (α = 0.88), and for subscales including negative affect (α = 0.96), challenge (α = 0.84), positive affect (α = 0.84), flow (α = 0.83), and immersion (α = 0.80); except for tension (α = 0.48). One item of ‘tension’ was therefore excluded, resulting in a Cronbach’s alpha of α = 0.55, showing poor yet acceptable internal consistency [56]. Negative affect and tension were reversed to create one overall positive score for the GEQ scale.

The GEQ’s competence subscale had been measured with another validated questionnaire on perceived competence, autonomy, and relatedness in serious games [57], including two items with good internal consistency for competence (α = 0.82), that have not been included in the overall GEQ score (Table 1).

To measure engagement in the narrative, three validated items on narrative involvement (α = 0.90) [58,59,60], and three validated items on narrative sensation (α = 0.92) [58,60] were used, with excellent internal consistency (α = 0.92) (Table 1). Finally, overall game appreciation was measured with an overall score ranging from 0 (not liking at all) to 10 (liking very much). The survey included an open-ended question where participants could leave their comments and suggestions for improvement of the game.

### 2.4. Analysis

Descriptive statistics were used to characterize both study conditions (i.e., gender, age, type of education, family affluence, gameplay frequency for digital games and exergames). To assess differences in baseline participant characteristics between conditions, chi-square tests (χ^2^) were conducted to assess differences between categorical variables and Mann–Whitney tests (*U*) were conducted to assess differences between continuous variables. A chi-square test assesses whether the distributions of categorical variables differ from each other: if what we observe in our data follows what we would expect if all data were distributed relatively equally across categories, our test would be non-significant, allowing us to conclude there is no difference in the distribution of participant characteristics between conditions. The Mann–Whitney U test assesses whether the median from one group is significantly different than from another group and is applied to independent samples (such as independent experimental conditions, as the case here).

Differences in engagement scores between conditions were assessed using Mann–Whitney tests, given the non-normal distribution of engagement variables based on a visual inspection of histogram, Q-Q plot and the Kolmogorov–Smirnov test in SPSS 25.0 (IBM Corp., Armonk, NY, USA). Level of significance was set at *p* < 0.05 [61]. Effect sizes were calculated for significant results [61]. Responses to the open-ended question were analyzed via inductive Thematic Analysis [62] in Nvivo software version 12 (QSR International Pty Ltd., Victoria, Australia) by one researcher (AD). A subset of these open-ended responses (10%) was independently double coded by another researcher (AS). The inter-rater reliability was considered high (κ = 0.85), differences were discussed until consensus was reached [63].

## 3. Results

### 3.1. Sample Description

Twenty of the 119 schools that were contacted participated (response rate 17%). Reasons for not participating in the study were participation in other research studies, time issues, or the aim of the research not aligning with their school policy (i.e., no focus on physical activity during that particular school year). In these schools, for 290 of 1544 adolescents both adolescent and parental consent was received, resulting in a study sample of 207 adolescents (response rate 13%). Adolescents may have declined because of eligibility reasons (i.e., not having an Android phone), or because they or their parents did not wish to consent to participate. Only those in the two intervention conditions for whom post-intervention measurement on user engagement was available, and who indicated to have played the game at least once were included in the quantitative analyses on subjective engagement scales of GEQ, competence and narrative engagement, as these questions measured their in-game experience and were not relevant to those who had not played the game (*n* = 94, mean age = 14.61 ± 1.93; 35% female). There were no significant differences between conditions on baseline characteristics (Table 2). Behavioral engagement was assessed for a subgroup (*n* = 59) of the eligible sample (*n* = 94), as complete and correctly exported game metric data to the game storage of number of sessions and total play time was only available for this subgroup. The analysis of open-ended questions comprised a subgroup (*n* = 42) of the eligible sample (*n* = 94) and included in addition comments by those who did not play the game (*n* = 8), as their first impressions could provide useful information on reasons for non-play.

### 3.2. Behavioral and Subjective Engagement

With regard to behavioral engagement (*n* = 59), play time data indicated a general low usage of the game. Median total play time was 8.70 min (interquartile range = 13.35) in the dynamically tailored condition and 24.77 min (interquartile range = 62.50) in the non-tailored condition, being significantly higher in the non-tailored condition (U = 251.00; *p* = 0.005; d = 0.45). It is interesting to note that in the dynamically tailored group, participant’s play time ranged from opening the game only once to 89.88 min of play time. In comparison, the non-tailored condition ranged from several seconds of play time to a total of 2404.38 min of play time. Accordingly, the non-tailored condition played a median amount of eight completed sessions (interquartile range = 305) compared to three completed sessions (interquartile range = 18) in the dynamically tailored condition (U = 233.50; *p* = 0.002; d = 0.57).

Results on subjective engagement (*n* = 94) indicated no differences between the dynamically tailored and the non-tailored group on any of the GEQ subjective game engagement measures (including negative and positive affect, challenge, flow, immersion, tension), on the competence scale, nor on the narrative involvement sub-scale. A significant difference was found for the narrative sensation sub-scale, which was higher in the dynamically tailored group, yet with a small effect (Table 3). The effect size for narrative sensation (presence) was small. The power associated with a small effect size of 0.38 was associated with a power of 1-β = 0.43, given the sample size of 92 participants (42 versus 52 participants) and an α level = 0.05 (calculated in G*power software). In general, engagement scores were moderate (Table 2). Overall, participants in the dynamically tailored group condition scored the game with an overall score of 7/10 points compared to participants of the non-tailored group who scored the game with 6/10 points (i.e., 0 indicating the lowest score; 10 indicating the highest score). Findings on game scores did not significantly differ between both intervention groups (U = 940.50; *p* = 0.40).

### 3.3. Participant’s Suggestions for Improvement

Twenty participants in the dynamically tailored condition provided comments and suggestions for improvement. Table 4 shows an overview and quotes of their comments. They felt the game lacked variation in activities, was too dark and not visually attractive, had technical problems and inconsistencies, took too long to play and lacked social interaction. Very few mentions were made about the smart T-shirt (*n* = 2), and these were both negative.

Thirty participants in the non-tailored condition provided comments and suggestions for improvement. Table 5 shows an overview and quotes of their comments. As in the dynamically tailored condition, they felt the game lacked variation in activities, was too dark and not visually attractive, had technical problems and inconsistencies, took too long to play and lacked social interaction. Slightly more often than in the dynamically tailored condition, participants mentioned they thought the game was fun. Interestingly, some participants mentioned that it was annoying you could cheat, which is exactly what the smart T-shirt in the other condition prevented them from doing.

## 4. Discussion

This study evaluated the role of dynamic tailoring in user engagement with a narrative-based serious exergame in a leisure setting. We expected to find higher subjective engagement on immersion, flow, enjoyment and presence, and higher behavioral engagement in terms of usage in the dynamically tailored group than in the non-tailored group. Our findings provided little support for this hypothesis, and showed that participants in the dynamically tailored group experienced a higher level of narrative sensation (presence), whereas no differences were found between conditions on flow, immersion, and enjoyment. Moreover, the non-tailored group played the game for longer and completed more sessions than the dynamically tailored group, showing higher levels of behavioral engagement. Several phenomena may explain these unexpected effects that are further detailed below. We first propose that a flooring effect and low initial engagement with the game overall may explain a lack of difference between conditions. Several potential reasons for low game engagement and suggestions for improvement are provided. Second, it is possible that the method for tailoring via a smart T-shirt was not sufficiently attractive to create a difference between the conditions and may have introduced additional technological and usability barriers to gameplay, such as failure to connect the game to the smart T-shirt. Finally, the difference between conditions in narrative sensation (presence) is elaborated on. The discussion concludes with some concrete recommendations.

### 4.1. Low Levels of Engagement

#### 4.1.1. Initial Game Engagement

First, behavioral engagement data showed that in both conditions, participants played the game only for a very brief time. Engagement is increasingly understood as a process in which a user cycles through different stages of initial engagement, periods of engagement, disengagement and reengagement [64,65]. Behavioral engagement data showed that several participants had already stopped playing the game after several seconds, which suggests that their initial game engagement was very low. Regular players of commercial exergames usually play for 50 min per bout [66]. In serious game research play time lasts approximately 30 min [67,68] to 60 min [69] when being instructed, yet is often not reported in studies conducted in leisure settings. Low initial engagement may have been due to the setting in which it was played and the game characteristics themselves. Low usage rates of games have been reported, especially in unstructured, leisure settings [26,70] where there is no built-in encouragement to play and where the game competes for time with other activities. Game design may need to be more encouraging when implemented in unstructured settings, such as providing booster activities, reminders and notifications [71], as was also suggested by some participants in this study, or by using social marketing activities such as promotion by influencers, offering demo videos, building a community, social media campaigns, or short humorous videos in which teachers or peers recommend the game, to promote the use of the game [72,73].

Moreover, in both study conditions, several comments were made by participants which indicate negative first perceptions of game characteristics, such as unattractive graphics and technical problems during starting up the game. Graphics in serious games for youth were indeed found to be important for affective engagement, interest, immersion and perceived usability of the game [74]. The graphic design and color scheme may have triggered a negative emotional reaction among respondents [73,75]. Graphics should raise interest by their level of novelty, but also reduce complexity to facilitate comprehension and meaningfulness [76]. Game design should include user checks on interest and perceived complexity, as perceptions may differ between designers and users. Indeed, people with artistic experience appear to prefer more complex designs than those without artistic expertise [76]. As one participant mentioned the graphics could have been simpler, and emotional response to graphic design may have hindered user engagement.

Technical problems were also a reason for not playing the game. Usability testing included in-house alpha testing and testing in a group of 16 adolescents in one session. Research indicates that game developers are increasingly faced with challenges relating to the complexity of designing games for multiple platforms (i.e., variety of available smartphone brands and models), which can result in technological constraints [73,77]. In our study, game testing is likely to have occurred to a similar limited and thus insufficient level, and have faced complexity challenges relating to multiple smartphone types, Android versions, and Bluetooth connections (relevant for the dynamically tailored condition), which are often reported as common problems in game design [73]. One study indicates that adolescents tended to report technical problems, while in fact they referred to usability issues such as misunderstanding instructions, or cumbersome activities [78]. Aligning definitions of technical problems and usability may help to better understand the particular problems in future studies. In general, technical problems are often associated with player frustration and demotivation to continue playing [73,79]. Extensive game testing in-house as well as in real-life settings is thus, for the indicated reasons, recommended. Such real-life settings should ideally reflect the context in which the individual is expected to use the intervention, and in which the intervention effects are expected to occur once implemented, to ensure ecological validity of study results. Future studies may even consider an integration of (semi)automated play testing and deep learning, reducing human resource testing and detecting technical issues in a timely manner [73].

#### 4.1.2. Continued Game Engagement

Research on what drives initial versus continued engagement in serious digital health games is to the best of our knowledge still lacking. Insights from continued engagement with commercial games mention hedonic factors of ease of use, novelty, challenge and esthetics [80], flow [81], sense of achievement [82], presence [83], utilitarian outcomes such as exercise and health or identity-related benefits [81,84,85], and social interaction or norms [81,82] are important drivers of continued game engagement. Some of these drivers also appeared in user comments, which could clarify why continued engagement was also low: the activities lacked variation, the missions took too long, the game experience was disappointing and lacked social interaction.

The game frequently mentioned ‘get to <narrative location>’ as an instruction, allowing adolescents to move freely, with no detailed instructions or suggestions for specific physical activity types (i.e., being free in choosing to run, jump, or climb). Without a clear instruction for the type of physical activity, this may have been interpreted as walking. Including specific instructions or suggestions for a variety of physical activity movements may be more engaging [86] for some players, as research outlines that different users either prefer [85] or are demotivated by specific instructions [85,87]. Variety in activities and missions is important in game-based learning as it nurtures individuals’ desire for stimulation and creates positive feelings via meeting curiosity. In fact, novelty has been suggested as an additional basic psychological need that complements perceived autonomy, competence and relatedness in the Self-Determination Theory, and an important predictor of motivation in physical activity education [88]. Novelty can also be considered as a contributor to perceived competence, as it provides new opportunities for competence satisfaction while using technology [89]. Additionally, in non-game based physical activity promotion, a variety in physical activities is encouraged to keep people motivated when improving their physical activity levels [90].

Users furthermore felt the missions took too long. The duration of a mission of around 10 min was on the one hand determined by a minimum 10-min bout of physical activity to gain health effects and on the other hand the average time to commute to school, during which adolescents had mentioned they would consider playing the game [53]. Recent insights, however, call into question that physical activity bouts need to last for a minimum of 10 min to count towards meeting the physical activity guideline [91]. It may therefore be more appropriate for future versions of the game to offer the player a choice of adaptable lengths of the missions [53,86].

A systematic review on user engagement with exergames furthermore showed that social interaction in the game can enhance engagement [92], as social comparison and exchanging social messages can contribute to enjoyment [93]. This was also suggested by a couple of respondents who would have preferred a multiplayer mode. The social interaction included in the SmartLife game had to be discovered via a lengthy process: after a while the player discovers they are not the only survivor and can interact with other players. It seems advisable to provide this feature as a visible element from the start, important for initial engagement. The fact that this game was only available for Android may have further prevented participants from playing the game, as they were not able to play with friends who owned a iOS iPhone and were not considered as eligible for this study.

### 4.2. Dynamic Tailoring

Second, dynamic tailoring did not appear to create a largely different subjective engagement between the conditions. We had especially expected to find a difference in the dimension of flow, which is created by an optimal balance between the challenge and a person’s capabilities. The sensor aimed to keep players engaged in sufficient levels of physical activity. This form of dynamic adjustment can be considered an automated form of feedback that is integrated in the narrative. Feedback has shown to be an important aspect of user engagement with exergames [94]. Integrating feedback into the game, as done here, rather than traditional, explicit feedback has been suggested as a way to decrease cognitive load and avoid being unpleasant and fatiguing [94]. However, from a psychological perspective, for feedback to be effective it should also allow comparison to others, past performances, and to goals, and strengthen a sense of mastery [94]. The integration of the sensor in the narrative and challenge may have provided a link to goals but may have insufficiently emphasized and reinforced their achievements. Apart from measuring and adapting, the game could have used the sensor information to provide more regular, encouraging audio-feedback to increase this sense of mastery and enhance the experience of flow. The open-ended comments provided by the participants indeed showed that the T-shirt with the sensor played a minor role in their game experience, and the very few mentions in the tailored condition referred to the negative experience of it being burdensome rather than useful [95].

The higher behavioral engagement in the non-tailored condition may have been caused by the smart T-shirt being considered a hassle to put on in the tailored condition or may have been an effect of being able to ‘cheat’ in the non-tailored condition, which was less present in the tailored condition. The integration of the sensor in a wearable device was considered a needed improvement over other available options that are often inaccurate, uncomfortable and allow cheating [71]. The comments in the non-tailored condition show that having tailoring in itself can indeed be useful, as they mentioned the fact that they could cheat as a weakness of the game. However, it may therefore be recommended to find other ways to include dynamic tailoring that do not require a smart T-shirt. In other mobile exergames, wearables are connected to the game [87], or an accelerometer smartphone app is used [85] to track the intensity of physical activity and contributed to enjoyment in the game [93]. Additionally, it may be possible that the provided T-shirt was perceived as not sufficiently fashionable [96], which could be overcome by providing personalization options for the smart T-shirt [53]. In future studies it may also be researched how dynamic tailoring can be applied to a broader range of game elements, not focusing only on physical activity challenges, but also on automatically modifying game types, preferences, play time, or even preferred game features [97] that can be released gradually to increase engagement [98].

### 4.3. Higher Narrative Presence

Finally, our findings did show a difference in narrative presence between the conditions. Participants in the dynamically tailored group felt they were more present in the narrative world than the actual world, compared to the non-tailored group. In this game, the narrative and challenge were adapted to the physical activity intensity levels of the player. This type of biofeedback can increase immersion and presence in the narrative. A game study that adapted the narrative of the game to player’s biofeedback found it could be engaging when the connection between the gameplay and biofeedback was clear, but was also considered as confusing, distracting and lowered engagement in users who found the link between biofeedback and gameplay less clear [99]. The commercial game ‘Zombies, Run!’ (Six to Star & Naomi Alderman; https://zombiesrungame.com (accessed on 6 May 2021); London, UK) also uses adaptation of the narrative by biofeedback to keep users more engaged. The mechanisms via which tailoring of the narrative based on biofeedback can affect narrative presence remain, to the best of our knowledge, unexplored. It is possible that biofeedback has created a stronger connection between the virtual world and personal life and has created a situation that approaches direct experience, which enhances presence [100]. It can also be expected to increase identification with characters, which is important for the strength of narrative persuasion [32]. This higher level of narrative presence in the tailored condition in our study, however, did not translate in a higher overall level of game engagement in this condition. Whereas presence is considered an important factor in game engagement by mechanisms of escapism, game engagement will only occur if the game does not only offer an escape from unpleasant real-life situations, but also offers an alternative that is more entertaining [101]. As users had many reservations about the general game design, these other design elements may need to be improved for narrative presence to have an effect on overall game engagement.

### 4.4. Strengths and Limitations

This study was limited by the low number of adolescents who played the game, which is often reported as a challenge in serious game research, especially in leisure time [68,102], and known to lack interest when played for several weeks [103]. An even lower sample for the behavioral engagement analysis was available, as not all game metric data was considered complete or not transferred to the game storage correctly. The effect size for narrative sensation (presence) was small. Internal consistency for one construct of engagement (tension) was low. The game type of an post-apocalyptic game was chosen in participatory design with the target group, but may not be the preferred game narrative for all adolescents. Future research may wish to offer several options to better suit the diversity in game story and type preferences. Strengths of this study are the study design of a cluster-randomized controlled trial to test dynamic tailoring as a mechanism of user engagement [37], thereby contributing to the limited experimental research on user engagement in serious games, especially exergames. Engagement was measured in a comprehensive manner using several indices, fitting the multidimensional nature of the concept. The study combined game metrics, quantitative analyses of scales and qualitative analyses of user comments to gain a richer understanding and insights into the findings. So far, to the best of our knowledge, no research is available on dynamic tailoring in narrative-driven exergames in leisure settings. Our study provided first insights and recommendations for future research and development.

## 5. Conclusions

In contrast to our hypotheses, our study did not find increased user engagement (except narrative presence) in the dynamically tailored version of the narrative-driven exergame in comparison to the non-dynamically tailored version. User comments highlighted several areas for improvement needed for the game itself to be more engaging, which were similar for both conditions. For initial engagement, simple and attractive graphics are needed. Technical bugs should be avoided by more extensive usability testing in real-life settings. Notifications and reminders should be provided to encourage users to return to the game. A large variety of activities that can be played in shorter or adaptable missions may keep users motivated to play. Social interaction in the game should be available as a basic component from the very start. Dynamic tailoring, however, did show potential in two areas. It created fewer possibilities to cheat, whereas the possibility to cheat in the non-tailored version was met with less respect or with not taking the game seriously. A different, less cumbersome or more attractive method for physical activity sensing might have created higher user engagement. Additionally, although dynamic tailoring did not improve flow, immersion or enjoyment, it did create a higher presence in the narrative world. As narratives are suggested as a way to further improve user engagement with exergames, the combination of narratives and attractive, unobtrusive ways of dynamic tailoring on biofeedback could hold potential for more engaging exergames, to be verified in future research.

## Figures and Tables

**Figure 1 ijerph-18-07444-f001:**
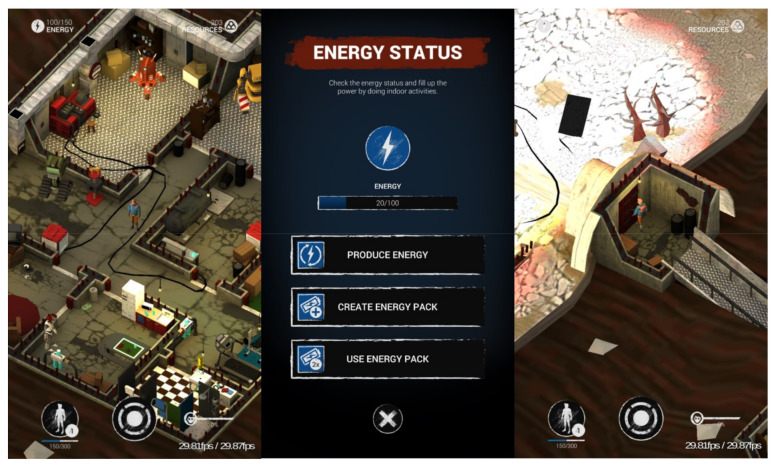
Impression of the game setting, similar in the dynamically tailored and non-tailored game version.

**Table 1 ijerph-18-07444-t001:** Definitions of included scales.

Scale	Definition
1. Game Engagement Questionnaire (GEQ)	
GEQ-Negative affect	Frustration and anger as a result of failing to reach a goal, as challenges are high and perceived as impossible to overcome, or boredom when a game is perceived as too easy for a player.
GEQ-Challenge	Optimally balanced challenged, i.e., reflected in goals or skills that need to be attained.
GEQ-Positive affect	Pride or euphoria as a result of fulfilment of a goal or mastery of a specific skill.
GEQ-Flow	State of optimal concentration and performance in the game as a result of the challenge of the game.
GEQ-Immersion	Perceiving the game world by means of graphics and the story, with some awareness of one’s surroundings.
GEQ-Tension	Experience of suspense and pressure in the game.
2. Competence	Mastery of a skill involves perceived competence.
3. Narrative engagement	
Narrative involvement	Feeling of being engrossed in a story.
Narrative sensation	Feeling of being there in the narrative world (i.e., presence).

**Table 2 ijerph-18-07444-t002:** Descriptive information on intervention groups.

Participant Characteristics	Dynamically Tailored (*n* = 42)	Non-Tailored (*n* = 52)	Significance Test
Gender	33% female	37% female	χ^2^ = 0.11, *p* = 0.75
Age	MDN = 14.00	MDN = 14.17	U = 1035.00; *p* = 0.67
Family affluence (low-medium/high)	57% high	40% high	χ^2^ = 2.62, *p* = 0.11
Education type (academic/non-academic track)	67% non-academic	60% non-academic	χ^2^ = 0.49, *p* = 0.48
Digital game frequency at baseline	55% plays daily	44% plays daily	χ^2^ = 1.03, *p* = 0.31
Exergame frequency at baseline	10% plays daily	8% plays daily	χ^2^ = 0.10, *p* = 0.75

**Table 3 ijerph-18-07444-t003:** Subjective user engagement.

Subjective Engagement	Dynamically Tailored (*n* = 42)	Non-Tailored (*n* = 52)	Mann–Whitney U Test
Game engagement (0–4)			
Overall GEQ ^1^	2.03	1.97	U = 1027.50; *p* = 0.62
GEQ-Immersion	2.00	1.67	U = 987.50; *p* = 0.42
GEQ-Positive affect	2.00	2.00	U = 1073.00; *p* = 0.88
GEQ-Flow	1.67	1.33	U = 901.50; *p* = 0.14
GEQ-Challenge	1.67	1.33	U = 1059.00; *p* = 0.80
GEQ-Negative affect	2.33	2.33	U = 1055.00; *p* = 0.78
GEQ-Tension	3.00	3.00	U = 1079.00; *p* = 0.92
Competence score (1–5)			
Competence	3.00	2.50	U = 882.50; *p* = 0.15
Narrative engagement (0–4)			
Overall narrative score	1.67	1.42	U = 923.00; *p* = 0.20
Narrative sensation	1.67	1.00	U = 817.50; *p* = 0.03 * d = 0.38 ^2^
Narrative involvement	2.00	2.00	U = 1064.50; *p* = 0.83

^1^ GEQ = Kids Game Experience Questionnaire score, including participants who played the game at least once, according to self-reported use and game metric data; * = Significance at 0.05 level; ^2^ d = effect size calculated according to G*power software (Heinrich Heine University Düsseldorf, Düsseldorf, Germany).

**Table 4 ijerph-18-07444-t004:** Comments provided by participants in the dynamically tailored condition.

Themes/Subthemes	Number of References	Quotes
**Activities**	11	“*I felt that you couldn’t do enough or sufficiently varied things in the bunker, the missions were sometimes monotonous and repetitive*” (male, non-academic education, playtime 49 min)
−lack of variation	4	
−insufficient number of activities	3	
−game did not motivate to move	2	
−missions too repetitive and monotonous	1	
+enjoyed the physical activity in game	1	
**General positive/negative affect**	10	“*It was very exciting and fun*” (male, non-academic education, playtime 11 min)
−not fun, stupid, not interesting	3	
−not stimulated to continue to play	2	
+fun game	4	
+exciting	1	
**Graphics and audio**	8	“*If the graphics had been better, I probably would have played it more often, the game was also quite dark and not colorful*” (male, non-academic education, playtime 0 min)
−poor quality graphics	3	
−too dark, not colorful enough	1	
−lack of visual support	1	
−irritating narrator voice	1	
+graphics reasonably OK	2	
**Technical problems**	7	“*The game connection would sometimes be lost and it took a very long time to get started*” (male, academic education, playtime 7 min)
−game did not work	3	
−too slow starting up	2	
−disconnecting problems	1	
−poor quality	1	
**Game experience**	6	A few negative points: *“<…> that this woman said she would go and turn on the music and eventually didn’t*” (male, non-academic education, playtime 31 min)
−inconsistencies in the game	2	
−no transfer to real-life, lacking realism	2	
−lack of perceived achievement in game	1	
−need more explanation	1	
**Time issues**	5	“*Often I had no time for it and when I did have time I often went to do something else*” (male, academic education, playtime 7 min)
−missions took too long to complete	3	
−lack of time	1	
−preferred to do other things with my time	1	
**Tailoring**	2	“*If the game is intended to get youth to move more, then I don’t think that many youngsters are going to bother to put on the T-shirt with the sensor*“ (female, non-academic education, playtime 0 min)
−smart T-shirt is too much of a bother to put on	2	
**Social interaction**	2	“*I’m more a console game who likes multiplayer games*” (male, non-academic education, playtime 0 min)
−prefer multiplayer games	2	

**Table 5 ijerph-18-07444-t005:** Comments provided by participants in the non-tailored condition.

Themes/Subthemes	Number of References	Quotes
**Activities**	13	“*Sometimes you did not have much to do or always had to do the same thing*” (female, academic education, playtime 33 min)
−insufficient number of activities	4	
−lack of variation	3	
−missions too repetitive and monotonous	3	
−game did not motivate to move	2	
+enjoyed the physical activity in game	1	
**General positive/negative affect**	13	“*I think this is really a fun game for people who want to move more!*” (female, academic education, playtime 69 min)
−not fun, stupid, not interesting	3	
−not stimulated to continue to play	2	
−in general not interested in playing mobile games	1	
+fun game	7	
**Game experience**	12	“*There were no encouragements from the game to play. I did not use it for quite a while and I did not get any notifications or so to play. It’s just a suggestion but if there was a notification at some point that I had not played yet, I would have been more motivated*” (female, academic education, playtime 17 min)
−inconsistencies in the game	4	
−need more explanation, unclear what to do	3	
−lack of notifications or encouragement to play	2	
−missing rewards in the game	1	
−no transfer to real-life, lacking realism	1	
−prefer first-person shooter games	1	
**Graphics and audio**	6	“*Graphics may have been better if they were a bit more simplistic, as it is they were not sufficiently attractive, maybe try another style?*” (male, non-academic education, playtime 0 min)
−poor quality graphics	5	
−too dark, not colorful enough	1	
**Technical problems**	6	“*During the sessions the voice and time did not match. During one of the sessions, the voice said the mission was done while I actually still had 6 min to go. This was quite bothersome during playing*” (female, academic education, playtime 17 min)
−game did not work	2	
−technical bugs	1	
−narration and content did not match	1	
−disconnecting problems	1	
−could not advance beyond first level	1	
**(Lack of) Tailoring**	3	“*You could put your phone down and start up the game while you yourself are lying down on the couch and you still have achieved your 10 out of 10 min of walking time*” (female, non-academic education, playtime 2404 min)
−ability to cheat on the physical activity duration	3	
**Time issues**	1	“*I would enjoy it more if the running did not take as long and if the movement missions would be more varied with other types of*” (female, academic education, playtime 29 min)
−missions took too long to complete	1	
**Social interaction**	1	“*Maybe also that you have to play with a friend would definitely help*” (male, academic education, playtime 19 min)
−prefer multiplayer games	1	

## Data Availability

The data presented in this study are available on request from the corresponding author. The data are not publicly available as they are still the subject of ongoing publications by the consortium.

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
