# Peer review of "Does Dynamic Tailoring of A Narrative-Driven Exergame Result in Higher User Engagement among Adolescents? Results from A Cluster-Randomized Controlled Trial"

_ijerph, 2021, doi:10.3390/ijerph18147444_

Round 1

Reviewer 1 Report

This is a honest paper in presenting an experiment whose results were not expected and, partly, are not consistent with the expected outputs. Nevertheless they represent, as such, a bright example of serendipity, in focussing the scholars attention on results of high interest, and which are worth investigating and taking into account for further applications. The results are carefully shown in terms of statistical analysis (chi square and Kolomogorov-Smirnov significance test); in this sense the only improvement i suggest is a synthetic explanation of the meaning of statistical models used, assuming that there could be a part of readers (human science scholars, above all) not confident with statistics. 

Author Response

Thank you for the positive feedback and for this suggestion. We have added the following information to the method section:

“To assess differences in baseline participant characteristics between conditions chi-square tests (χ²) were conducted to assess differences between categorical variables and Mann-Whitney tests (U) were conducted to assess differences between continuous variables. A chi-square test assesses whether the distributions of categorical variables differ from each other: if what we observe in our data follows what we would expect if all data were distributed relatively equally across categories, our test would be non-significant allowing us to conclude there is no difference in the distribution of participant characteristics between conditions. The Mann–Whitney U test assesses whether the median from one group is significantly different than from another group and is applied to independent samples (such as independent experimental conditions, as the case here).”

Reviewer 2 Report

The work aimed to evaluate whether dynamic tailoring in a narrative-driven mobile exergame for adolescents played in leisure settings, can create higher user engagement compared to a non-tailored exergame. Results  show that user engagement was low in both conditions and the narrative sensation was higher in the dynamically tailored condition, but the non-tailored condition showed longer play-time. Validation was carried out with an accelerometer sensor in a T-shirt that connects to the smartphone via Bluetooth.

The study is very useful for future research and development in this topic.

The topic fits very well the scope of the journal. 

The manuscript is well written, the structure of the paper is clear and the language is proper. 

The contributions should be well delimited in the introduction section in order to clarify regarding state of the art related solutions. Here, the focus should be similar tentative solutions that use smart devices (such wearables) as the proposal uses an accelerometer.

In this regard, why using only accelerometer? There are other devices with many types of sensors that could help and improve the smart wearables (in this case smart T-shirt). The authors should clarify this justification and context in the manuscript.

The last paragraph of section 1 should write the organisation and structure of the manuscript.

The discussion section goes in deep enough in order to understand the results obtained. However, I recommend better organisation of the section such as dividing in subsections regarding the sub topic results. It could help the reader. 

Author Response

The work aimed to evaluate whether dynamic tailoring in a narrative-driven mobile exergame for adolescents played in leisure settings, can create higher user engagement compared to a non-tailored exergame. Results  show that user engagement was low in both conditions and the narrative sensation was higher in the dynamically tailored condition, but the non-tailored condition showed longer play-time. Validation was carried out with an accelerometer sensor in a T-shirt that connects to the smartphone via Bluetooth. The study is very useful for future research and development in this topic. The topic fits very well the scope of the journal. The manuscript is well written, the structure of the paper is clear and the language is proper. 

Thank you for this positive feedback!

The contributions should be well delimited in the introduction section in order to clarify regarding state of the art related solutions. Here, the focus should be similar tentative solutions that use smart devices (such wearables) as the proposal uses an accelerometer. In this regard, why using only accelerometer? There are other devices with many types of sensors that could help and improve the smart wearables (in this case smart T-shirt). The authors should clarify this justification and context in the manuscript.

Thank you for this suggestion. We have added the following information to the introduction:

“The biofeedback was enabled by a flat and small sensor (accelerometer) placed in a 3x3cm pocket at the back of a T-shirt specifically designed for the purpose of this study. A T-shirt may offer comfort and the potential for personalization in different colors and prints, which may appeal to an adolescent target population. Battery life and washability are important issues as well that can present some manufacturing challenges for smart shirts. The choice for a sensor in the shirt rather than a fully embedded sensor in the fabric was driven by reasons of feasibility in this stage of the intervention's pilot testing (46). Ideally in a finalized product, the measurement would be made by conductive textile, where the sensor is not immediately visible and that appears more natural and integrated to the user. As the purpose of this paper was to assess the role of the biofeedback in user engagement, a fully embedded sensor was not considered needed here. Despite some increasing interest in such solutions (see e.g. 47 for a study protocol of an intervention combining a smart shirt and serious game), we are not aware of any other study assessing the effects of the combination of a smart shirt and a serious game on user engagement, which again highlights the innovativeness of this study. “

(46) Angelucci, A., Cavicchioli, M., Cintorrino, I. A., Lauricella, G., Rossi, C., Strati, S., & Aliverti, A. (2021). Smart textiles and sensorized garments for physiological monitoring: A review of available solutions and techniques. Sensors21(3), 814.

(47) Puigdomenech, E., Martin, A., Lang, A., Adorni, F., Gomez, S. F., McKinstry, B., ... & Espallargues, M. (2019). Promoting healthy teenage behaviour across three European countries through the use of a novel smartphone technology platform, PEGASO fit for future: study protocol of a quasi-experimental, controlled, multi-Centre trial. BMC medical informatics and decision making19(1), 1-13.

The last paragraph of section 1 should write the organisation and structure of the manuscript.

Thank you for this suggestion, we have added the following information at the end of the introduction:

“This paper first describes the intervention, methods of data collection and analyses; next presents the characteristics of the analyzed sample, results of the quantitative analyses on engagement, and of qualitative analyses of user comments, to end with a discussion of these results, their contribution to literature and practical implications. Some limitations and strengths are provided as a final note to the paper.“ 

The discussion section goes in deep enough in order to understand the results obtained. However, I recommend better organisation of the section such as dividing in subsections regarding the sub topic results. It could help the reader. 

Thank you for this useful suggestion! We have added the following subtitles to structure the discussion in subsection:

4.1. Low levels of engagement; 4.1.1 Initial game engagement; 4.1.2. Continued game engagement;

4.2. Dynamic tailoring;

4.3. Higher narrative presence;

4.4. Strengths and limitations

Reviewer 3 Report

this paper is easy to read thru and to follow in reasoning. they tried to tackle the low user engagement, however, they did not really succeed in this. They do report ways to continue tackling this. In that sense, it is a good contribution. I do not have much to say about it. It is clear that this paper has been revised and issues adressed. 
I think that the number of keywords are perhaps too many, they are 8. six would be preferable. Perhaps health intervention (as this do not measure health outcomes) and dynamic difficulty adjustment (as this is not either the main focus) could be dropped from keywords. But this is only a suggestion. 

When it comes to measurement, I had to flip back and forth in order to understand all scales, as scale names are given in table 1, but not as easily identified in the text. 

Subjective engagement is the name of the scale in the text in user engagement, but overall GEQ in table 3. Finding subjective engagement and identifying overal GEQ on the other end, indicating a flipping between pages. Using the same term, and perhaps italics them in tezt so they stand out a bit when you want to understand what is measured. 

The text on user engagement (2.3.2) is condence and not really reader friendly. Looking at table 1, these seem to be the scales used, but these are subscales of the overal GEQ scale (i.e. subjective engagement). So table 1 are giving definitions of the subscales of the GEQ and including other scales. 

I think that you could make both the explanation in table 1 clearer and what is measured by GEQ and by other measures. I do not have any good solution for this, perhaps to add a couple of more columns to table 1, perhaps put in the number of items and the alpha of these there. create a new paragraph for instruments that are not GEQ. 

Author Response

this paper is easy to read thru and to follow in reasoning. they tried to tackle the low user engagement, however, they did not really succeed in this. They do report ways to continue tackling this. In that sense, it is a good contribution. I do not have much to say about it. It is clear that this paper has been revised and issues adressed. 

Many thanks for this positive feedback to our paper.

I think that the number of keywords are perhaps too many, they are 8. six would be preferable. Perhaps health intervention (as this do not measure health outcomes) and dynamic difficulty adjustment (as this is not either the main focus) could be dropped from keywords. But this is only a suggestion. 

Thank you for this suggestion. We have dropped the keyword 'health intervention' as this is already covered under the more specific term of serious game but have decided to retain dynamic difficulty adjustment as the tailoring we provided does dynamically adjust the difficulty of the challenge, and the paper may be interesting for authors who wish to investigate this topic. We prefer to have one additional keyword rather than the paper being missed by interested readers.

When it comes to measurement, I had to flip back and forth in order to understand all scales, as scale names are given in table 1, but not as easily identified in the text. Subjective engagement is the name of the scale in the text in user engagement, but overall GEQ in table 3. Finding subjective engagement and identifying overal GEQ on the other end, indicating a flipping between pages. Using the same term, and perhaps italics them in tezt so they stand out a bit when you want to understand what is measured. 

Thank you for pointing out that this was not sufficiently clear. We have harmonized the terms of the scales and subscales between the text and tables, so that all scale names are the same in the text and tables. This has been changed as following:

  • “Results on subjective engagement (n=94) indicated no differences between the dynamically tailored and the non-tailored group on any of the GEQ subjective game engagement measures (including negative and positive affect, challenge, flow, immersion, tension), on the competence scale, nor on the narrative involvement sub-scale. A significant difference was found for the narrative sensation sub-scale, which was higher in the dynamically tailored group, yet indicating a small effect”
  • Tables 1 and 3 now mention GEQ prior to the names of the sub-scales of GEQ

The text on user engagement (2.3.2) is condence and not really reader friendly. Looking at table 1, these seem to be the scales used, but these are subscales of the overal GEQ scale (i.e. subjective engagement). So table 1 are giving definitions of the subscales of the GEQ and including other scales. I think that you could make both the explanation in table 1 clearer and what is measured by GEQ and by other measures. I do not have any good solution for this, perhaps to add a couple of more columns to table 1, perhaps put in the number of items and the alpha of these there. create a new paragraph for instruments that are not GEQ. 

Thank you for this suggestion. We have added an overarching title (e.g. ‘1. Game Engagement Questionnaire’, ‘2. Competence’, ‘3. Narrative engagement’) and numbers for the three main scales and have used an indent in the table to show GEQ subscales, as well as having mentioned ‘GEQ’ before each GEQ subscales. In the text, we have started a new paragraph for the scales that were not part of GEQ.

Reviewer 4 Report

IJERPH-1229786 presents results related to gaming from a cluster-randomized trial of adolescents. While some parts of this manuscript were interesting, other areas could be improved. I hope the authors consider my feedback.

MAJOR COMMENTS

  • Introduction: This section is far too verbose. As a reader, I lost interest because the Introduction just ran-on. Strongly consider reducing text where appropriate for being more to the point. The same comment could be applicable to the Discussion section.
  • Lines 112-131: This manuscript looks to be a resubmitted version and I am a new reviewer. These lines of text appear to have been added from the previous version because they are highlighted in yellow, but this reviewer feels like this text is out of place relative to the Introduction section.
  • Results: The use of mixed methodologies does not seem to corroborate with the research overall. The subjective user engagement does not link with commentary regarding the themes of gaming. There just seems to be a disconnect in the purpose and methods that has not been well explained to this point.
  • Limitations: A different game aside from something post-apocalyptic may have connected better with some participants than others.
  • This reviewer is having a hard time with finding the significance of the study. What might be more significant is presenting how these games increase physical activity levels and long-term participation.

MINOR COMMENTS

  • Section 2.1: More information about how the convenience sample was recruited is warranted.
  • Lines 185-189: Please justify dichotomizing this item when there were seven categories.
  • Line 495: Avoid presenting results in a Discussion section.

Author Response

IJERPH-1229786 presents results related to gaming from a cluster-randomized trial of adolescents. While some parts of this manuscript were interesting, other areas could be improved. I hope the authors consider my feedback.

Thank you for this positive feedback and suggestions.

MAJOR COMMENTS

  • Introduction: This section is far too verbose. As a reader, I lost interest because the Introduction just ran-on. Strongly consider reducing text where appropriate for being more to the point. The same comment could be applicable to the Discussion section.

Thank you for this comment. We have checked the introduction again, which was 1.5 pages long and unfortunately could not find any element that could be reduced without missing crucial information to frame the study we conducted. Moreover, from other comments in this review, we had the impression that certain points of justification were not sufficiently clear in the introduction and needed expansion rather than reduction. We could therefore unfortunately not reduce the text without losing the theoretical foundations and justification of the novelty of this study.

We did however try to improve readability of this section by adding subtitles that indicate the structure of the text and hopefully aid in better retaining the reader’s attention.

  • Lines 112-131: This manuscript looks to be a resubmitted version and I am a new reviewer. These lines of text appear to have been added from the previous version because they are highlighted in yellow, but this reviewer feels like this text is out of place relative to the Introduction section.

Indeed, this information was added at the request of another reviewer in a previous round, who has already accepted the manuscript in its current form based on the revisions. As we assume the acceptance is conditional to his/her comments having been addressed, we feel it would not be correct to remove this text at this point. The reviewer may not have agreed to the manuscript without this text in place.

  • Results: The use of mixed methodologies does not seem to corroborate with the research overall. The subjective user engagement does not link with commentary regarding the themes of gaming. There just seems to be a disconnect in the purpose and methods that has not been well explained to this point.

Thank you for pointing out that the added value of the mixed methods approach was not sufficiently explained in the introduction and method section. We have now more explicitly referred to guidelines on how to measure and understand user engagement with digital health interventions in the method section, that report on the benefits of using validated scales in combination with more qualitative approaches, as done in our study. We have referred to some sources for the interested reader, but have not expanded in detail on this, to not conflict with the first comment of trying to reduce the text. The following information has been added:

“A mixed-method approach of quantitative analysis of scales and qualitative analysis of open-ended responses was chosen to combine their advantages in measuring engagement. Using standardized scales allows comparisons by user characteristics but lacks depth, whereas a qualitative analysis of individual’s experiences is time-consuming to analyze, requires participants to understand their motives and behavior, but offers more insights and potential hypotheses to understand trends in quantitative data (Yardley et al., 2016; Short et al., 2018)“

  • Limitations: A different game aside from something post-apocalyptic may have connected better with some participants than others.

Indeed, this choice was based on participatory research with the target group, but we can never be sure that our qualitative, participatory approach provided a game type to everybody’s liking. We have added the comment to the limitation section that different preferences exist for games depending on user characteristics. The following information was added to the limitation section:

“The game type of a post-apocalyptic game was chosen in participatory design with the target group, but may not be the preferred game narrative for all adolescents. Future research may wish to offer several options to better suit the diversity in game story and type preferences. “

  • This reviewer is having a hard time with finding the significance of the study. What might be more significant is presenting how these games increase physical activity levels and long-term participation.

We regret that we have not been able to sufficiently explain the relevance of initial and continued engagement in digital health interventions. For this reason, we have chosen not to reduce the text on the importance of engagement in the introduction. The following information was added to highlight the significance of understanding user engagement:

"Engagement is a precondition for health interventions to be effective: if interventions are not used, or not in the way, as long or as intensively as is needed, we cannot expect these interventions to have an effect on behavioural or clinical outcomes [10, 24]. Engagement is influenced by characteristics of the intervention (e.g. content, technological features) and in its turn influences determinants of behaviour change by the depth of involvement with the intervention: an intervention that is better appreciated by users will yield higher attention and motivation to interact with the intervention components that create effects [24, 25]. Understanding game engagement and what contributes to it, is thus important for digital health interventions to have an impact on public health."        

MINOR COMMENTS

  • Section 2.1: More information about how the convenience sample was recruited is warranted.

Thank you for this comment. The following information has been added:

“The schools were contacted via the personal network of the researchers (e.g. schools where master thesis students who were helping out with recruitment were alumni, that were located in the vicinity of the university, of which the researchers or colleagues knew some of the staff).”

  • Lines 185-189: Please justify dichotomizing this item when there were seven categories.

The use of seven categories for frequency of game play was inspired by research that has often been conducted among game players, or even among problematic game players. The scale did not show a normal distribution in our general adolescent population sample, which comprised many adolescents who were not regular game players. Based on this, we decided the 7-category scale was too granular for our sample and recoded into two categories. This justification has now been added to the paper as following:

“As this variable was severely skewed, it was dichotomized as playing daily (≥daily) and playing infrequently (<daily).”

  • Line 495: Avoid presenting results in a Discussion section.

Thank you for this comment. We have moved the mention of the power analysis to the result section and removed the power analysis results from the discussion section

Round 2

Reviewer 1 Report

I seems to me all the raccomandations have been adequately followed.

Author Response

Thank you for this positive feedback. We are glad to hear that our previous responses met your recommendations and that you have no further comments to our paper.

Reviewer 2 Report

Authors have improved the manuscript and I accept it in the present form.

Author Response

Thank you for this positive feedback. We are glad to hear that our previous responses met your recommendations and that you accept our paper in its current format.

Reviewer 3 Report

They have addressed the comments that I had. 

Reviewer 4 Report

The authors have address the previous concerns of this reviewer.